# ShuffleNet v2.3-StackedBiLSTM-Based Tool Wear Recognition Model for Turbine Disc Fir-Tree Slot Broaching

**Shenshun Ying [1,\*], Yicheng Sun [1], Fuhua Zhou [1] and Lvgao Lin [2]**

[1] Key Laboratory of Special Purpose Equipment and Advanced Processing Technology, Zhejiang University of Technology, Hangzhou 310023, China
[2] Zhejiang CHR Intelligent Equipment Co., Ltd., Lishui 321404, China
\* Correspondence: yss@zjut.edu.cn

**Abstract:** At present, deep learning technology shows great market potential in broaching tool wear state recognition based on vibration signals. However, traditional single neural network structure is difficult to extract a variety of different features simultaneously and has low robustness, so the accuracy of wear status recognition is not high. In view of the above problems, a broaching tool wear recognition model based on ShuffleNet v2.3-StackedBiLSTM is proposed in this paper. The model integrates ShuffleNet v2.3, which has been channel shuffling, and StackedBiLSTM, a long and short-term memory network, to effectively extract spatial and temporal features for tool wear state recognition. Based on the innovative recognition model, the turbine disc fir-tree slot broaching experiment is designed, and the performance index system based on confusion matrix is adopted. The experimental research and results show that the model has outstanding accuracy, precision, recall, and F1 value, and the accuracy rate reaches 99.37%, which is significantly better than ShuffleNet v2.3 and StackedBiLSTM models. The recognition speed of a single sample was improved to 8.67 ms, which is 90.32% less than that of the StackedBiLSTM model.

**Keywords:** deep learning; vibration signal characteristics; broaching; channel shuffling; tool state recognition



## 1. Introduction

Turbine disk is the most important rotating part of aero-engine, and its performance is related to the safety and reliability of various high-end equipment. The fir-tree slot joint is widely used in the installation of turbine disc and blade due to the advantages of less processing materials and convenient disassembly and assembly. The fir-tree slot joint has high machining precision, complex structure, and strict surface quality requirements [1]. Broaching is usually used to improve the efficiency of fir-tree slot processing. The fir-tree slot broaches have calibration part and finishing edges in addition to the roughing, semi-finishing, and finishing segments, with excellent surface finish on the workpiece. If broach damage (wear, tooth breakage, etc.) occurs during machining, it will scratch the workpiece and lead to additional manufacturing cost [2]. Therefore, in order to ensure the machining quality of the workpiece and the stability of the running process of the machine tool, the intelligent monitoring of the tool wear state of the machine tool has become a research hotspot in recent years, which has been widely concerned by the industry.

The wear failure and remaining useful life (RUL) of turbine disc fir-tree slot broach of aero-engine have attracted much attention in recent years. The monitoring of tool wear can generally be divided into direct monitoring method and indirect monitoring method [3]. However, direct monitoring has obvious drawbacks. First, it is necessary to stop the machine for inspection. Second, it is impossible to estimate the sudden damage during processing, which limits its application. Therefore, indirect monitoring method has become the research focus. In recent years, researchers in various countries have used various sensors to collect various process monitoring signals (such as vibration [4,5], acoustic

emission (AE) [6], cutting force [7], current and power [8], etc.) and applied support vector machines (SVM) [9,10], random forest (RF), hidden Markov algorithms [11,12] and Naive Bayesian [13,14] algorithms and other methods to monitor tool wear status and life. Machine learning has become a major research direction for tool condition monitoring with its advantages of efficiency and high accuracy. Axinte et al. [15] used sensors to extract effective perception signals and used them for training and testing probabilistic neural networks (PNN). Broaching test results show that the states of broaching tool such as wear, fracture and chipping can be automatically classified by the network. Kong et al. [16,17] proposed a tool wear estimation model based on Gaussian Mixed Hidden Markov Model (GMHMM) and Hidden Semi-Markov Model (HSMM). Based on verification, GMHMM tool wear state recognition model can effectively recognize tool wear state, and is superior to back propagation neural network (BPNN) model in accuracy and stability, which also lays a foundation for future industrial applications. Shi et al. [18] combined the least square support vector machine (LS-SVM) and principal component analysis (PCA) technology to propose a broaching tool wear state prediction model. Broaching experiments show that principal component analysis uses multi-sensor fusion technology to dig deeper into the characteristics related to tool wear. In addition, the tool wear prediction value constructed by LS-SVM is in good agreement with the tool wear value measured by optical scanning microscope. Axinte et al. [19] studied broaching tools under various conditions, namely freshly ground teeth, chipped teeth, weakened teeth, and overall uniformly worn cutters. The broaching experiments show that different sensor signals in the time or frequency domain have different recognition effects on the tool condition. And the cutting force signal, AE signal, and vibration signal are very sensitive to the change of tool state. Although some of the above research methods using machine learning have yielded good results in some areas. However, machine learning is difficult to model nonlinear data correlation polynomial regression, and heavily depend on the type of data and learning models. Second, machine learning is difficult to express highly complex data, and thus not easily generalizable.

With the continuous development of intelligent manufacturing, deep learning has gained widespread attention in the field of machine tool monitoring using excellent adaptive feature learning capability and excellent portability. Deep learning algorithms such as convolutional neural networks (CNN) [20,21], sparse autoencoder, recurrent neural networks (RNN), and deep belief networks (DBN) [22] have made significant progress in the field of tool wear and life prediction. Ma et al. [23] established a tool wear prediction model based on convolutional bidirectional long short-term memory (LSTM) network for milling force signals. The results show that the errors of the predicted values are all within 8%, which provides a new method for on-line monitoring of tool wear. Li et al. [24] proposed a deep CNN model for predicting the remaining life of equipment adopting a time window approach to extract features, and achieved good results and has good advantages. Kothuru et al. [25] established a depth model of tool condition monitoring based on CNN to monitor tool wear by analyzing the spectral features of sound signals in the machining process. Finally, the test and visual analysis are carried out to lay the foundation for industrial application. Chen et al. [22] established a four-layer DBN to train the time-domain characteristics and calibrated tool wear degree of various signal data sets collected, and compared them with support vector regression (SVR) and artificial neural network (ANN) models. The results show that the training time of SVR model is relatively long, and the stability of ANN model is poor, while the accuracy and stability of DBN model is the best, and the running time is short. Although the tool wear state recognition technology based on deep learning has received widespread attention, it rarely involves considering the multi-scale features of sensor signals (i.e., the multi-dimension of signals, such as space and timing), which leads to incomplete feature extraction. In addition, most of the above studies focus on turning, milling processing. The tool wear monitoring technology based on deep learning for fir-tree slot broaching is still in the verification stage both theoretically and technically. There is still a long way to go to recognize the state of

broaching tool in different broaching stages, where the cutting edges are distributed in different positions and directions.

This paper aims at the problems of recognition accuracy and speed of wear state caused by complex geometric shape and different spatial distribution of existing fir-tree slot broaching tools. A new method of tool wear state model based on ShuffleNet v2.3 and StackedBiLSTM is proposed. Based on the existing theory, it can simultaneously extract spatial features and time series data, which solves some problems of single recognition and low accuracy in the current tool wear state methods. The main chapters of this paper are arranged as follows: The introduction section reviews the current the research status of cutting tool wear state recognition and conventional method. The second section proposes a new deep learning model, which combines ShuffleNet v2.3 using channel shuffling and long short term memory network to obtain rich wear signal features. In the third section, the ShuffleNet v2.3-StackedBiLSTM is verified by the turbine disc fir-tree slot broach machining experimental platform. The experimental results show that ShuffleNet v2.3-StackedBiLSTM has better accuracy, precision, recall and recognition speed.

## 2. Tool Wear Recognition Method Theory of Fir-Tree Slot Broaching

### 2.1. Spatial Feature Extraction Based on ShuffleNet v2.3

In high performance networks such as ResNeXt [26,27] and MobileNet [28,29], $1 \times 1$ pointwise convolution occupy a large amount of computational resources. Based on this problem, ShuffleNet [30,31] uses channel shuffling operation to effectively reduce the calculation of $1 \times 1$ pointwise convolution and fuses the information between channels, which makes it an extremely efficient lightweight network.

Convolution layer and pooling layer usually can deepen the network by stacking, which can greatly promote the recognition accuracy. However, as the number of layers in a neural network increase, on the one hand the problem of gradient disappearance becomes more apparent and gradient updates decay exponentially. On the other hand, the model is over-fitted, which makes the model less accurate. However, ResNet enhances the flow of information between the front and back layers by 'shortcut' the connections between the front and back layers, which to a certain extent alleviates the gradient disappearance phenomenon to some extent [32,33].

ShuffleNet v2 further reduces the amount of computation on the basis of clever using channel shuffling and 'shortcut'. The model makes the best in the balance of speed and accuracy, performs far better than networks like ResNet and Xception, and is very suitable for the application of mobile models.

Drawing on the ShuffleNet v2 neural network, the ShuffleNet v2.3 network structure is designed in this paper as an adaptive feature extractor for tool states. Compared with the original ShuffleNet v2-0.5x, the network has been significantly adjusted as shown in Table 1. Compared to ShuffleNet v2-0.5x, ShuffleNet v2.3 has changed the number of block modules in the 3 stages to a total of 13. The number of convolutional channels in the block modules has been adjusted. The maximum number of convolutional output channels has changed from 192 to 144. The more channels in the convolutional layer, the richer the features that can be learned, but the larger the number of parameters and the size of the model. Therefore, the number of convolutional channels should be reduced when it is possible to achieve the required accuracy of the model output. At the same time, part of the ShuffleNet v2.3 structure undergoes depthwise separable convolution, the obtained features correspond to low-dimensional space with fewer features, which makes the effect of the model worse. To address this problem, ShuffleNet v2.3 directly removes the last ReLU in each block, reducing the loss of features and obtaining better accuracy. Finally, dropout is used to prevent overfitting.

Figure 1 shows the ShuffleNet v2.3 network structure. In the ShuffleNet v2.3 network structure, each block module contains at least three convolutional layers and three Batch Normalization, 1 ReLU activation function. Among them, the $1 \times 1$ convolutional layer is used to reduce the number of channel dimensions and reduce the number of parameters,

increasing the nonlinearity and the ability to interact with information across channels, thus improving the expressiveness of the network. The batch normalization layer is used to solve the problem of internal covariate shifts during network updates.

**Table 1.** Comparison of ShuffleNet v2-0.5x and ShuffleNet v2.3 parameters.

| ShuffleNet v2-0.5x | Ksize | Stride | Repeat | Output Channels | ShuffleNet v2.3 | Ksize | Stride | Repeat | Output Channels |
|---|---|---|---|---|---|---|---|---|---|
| Conv1d | 3 | 2 | 1 | 24 | Conv1d | 3 | 2 | 1 | 24 |
| BatchNorm1d | - | - | 1 | 24 | BatchNorm1d | - | - | 1 | 24 |
| Maxpool | 3 | 2 | 1 | 24 | Maxpool | 3 | 2 | 1 | 24 |
| Stage2 | 3,1;3,1;1,3,1 | 2 | 1 | 48 | Stage2 | 3,1;3,1;1,3,1 | 2 | 1 | 36 |
| Stage2 | | 1 | 3 | | Stage2 | | 1 | 2 | |
| Stage3 | 3,1;3,1;1,3,1 | 2 | 1 | 96 | Stage3 | 3,1;3,1;1,3,1 | 2 | 1 | 72 |
| Stage3 | | 1 | 7 | | Stage3 | | 1 | 6 | |
| Stage4 | 3,1;3,1;1,3,1 | 2 | 1 | 192 | Stage4 | 3,1;3,1;1,3,1 | 2 | 1 | 144 |
| Stage4 | | 1 | 3 | | Stage4 | | 1 | 2 | |
| Conv1d | 1 | 1 | - | 1024 | Conv1d | 1 | 1 | - | 576 |
| GlobalAvgPool | 7 | 1 | - | 1024 | AvgPool1d | 3 | 1 | - | 576 |
| FC | - | - | - | 1000 | Dropout | - | - | - | 576 |
| Total params | | | 335,168 | | Total params | | | 148,182 | |
| Total mult-adds(M) | | | 378.59 | | Total mult-adds(M) | | | 173.06 | |
| Params size(MB) | | | 1.34 | | Params size(MB) | | | 0.59 | |
| Estimated Total | | | 104.92 | | Estimated Total | | | 67.56 | |

Note: Stride = 2 for the $3 \times 3$ convolution of the first block of each Stage, and Stride = 1 for the $3 \times 3$ convolution of the rest of the blocks, and repeat.

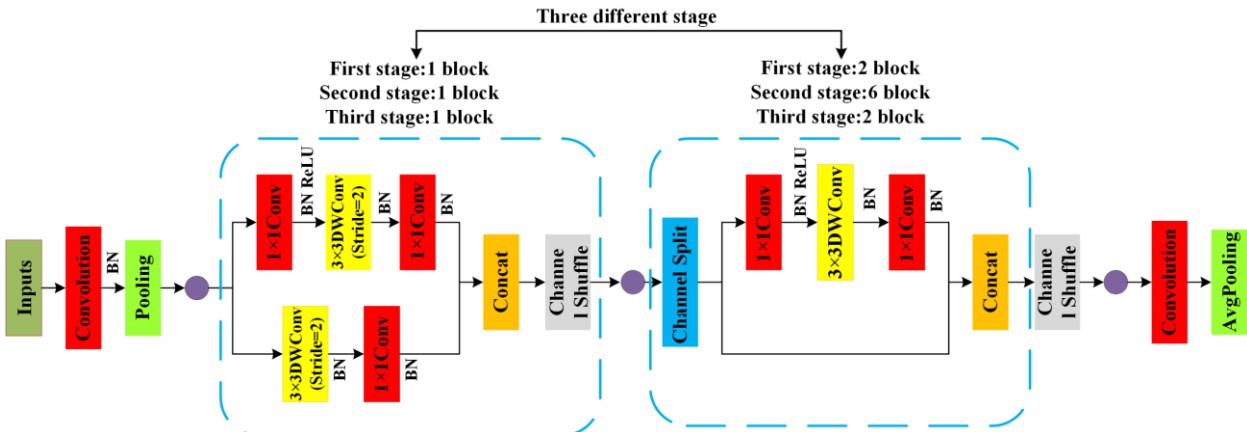

**Figure 1.** ShuffleNet v2.3 network structure.

### 2.2. Temporal Feature Extraction Based on StackedBiLSTM Unit Network

The conventional recurrent neural network RNN mainly uses historical information to assist in current decision making, and in theory it can remember the information seen in many time steps ago, but in practice it cannot obtain such long-term dependence information. The reason for this lies in problems such as gradient disappearance, gradient explosion. In order to solve problems such as gradient disappearance, gradient explosion, and so on, LSTM [34] networks are often used. The LSTM model structure not only captures longer dependencies, but also has better representation performance and better learning ability for neural networks with larger data sets.

The LSTM has the same inputs and outputs as the RNN, but the special feature of the LSTM is the introduction of gating units. For each moment $t$, the LSTM is divided into a total of three gating units such as input gate $i_t$, forget gate $f_t$, output gate $o_t$, etc. The specific formula is as follows:

$$i_t = \sigma(W_i \cdot [h_{t-1}, x_t] + b_i) \tag{1}$$

$$f_t = \sigma(W_f \cdot [h_{t-1}, x_t] + b_f) \tag{2}$$

$$\widetilde{C}_t = \tanh(W_c \cdot [h_{t-1}, x_t] + b_c) \tag{3}$$

$$C_t = f_t \cdot C_{t-1} + i_t \cdot \widetilde{C}_t \tag{4}$$

$$o_t = \sigma(W_o \cdot [h_{t-1}, x_t] + b_o) \tag{5}$$

$$h_t = o_t \cdot \tanh(C_t) \tag{6}$$

where $\sigma$ is a sigmoid function that specifies the value of the update gate between 0 and 1. $W_i$, $W_f$, $W_c$, $W_o$ represent the cyclic weight matrix of the LSTM unit, $b_i$, $b_f$, $b_c$, $b_o$ represent the bias of the LSTM unit. $h_{t-1}$ represents the previous moment hidden layer state, $x_t$ represents the current moment input, $\widetilde{C}_t$ represents the temporary unit state, and $C_t$ represents the unit state at the current time.

The RNN not only needs to forget the previous part of the memory, but also needs to input the latest memory, which is determined by the input gate. The forget gate jointly decides how much historical information to retain based on the current time input, the output of the previous moment, and the bias term of the forget gate. The output gate determines the output value by sigmoid function and tanh function according to the unit state.

The single-layer LSTM neural networks are weak in extracting data features and inadequate in representation capability. In this paper, we design stacking multiple LSTM layers to form a deep LSTM neural network, which enhances the feature representation capability of the neural network model and makes the prediction of the network more accurate. However, in the tool wear state prediction, the output of the current moment depends not only on the information of the previous moment, but also relates to the information of the subsequent moment. The bidirectional long and short term memory neural network contains two LSTM network layers which can capture both forward and reverse dependencies, thus learning more features of the temporal data. Therefore, in this paper, a bidirectional LSTM model is used to enhance the network capability and obtain a richer feature representation.

Existing deep learning frameworks can arbitrarily build neural network models and fuse the above two LSTM models to obtain a stacked bidirectional LSTM model, which integrally improves the representation capability of the network.

So far, this paper has proposed a new model for broaching tool wear recognition based on ShuffleNet v2.3 and StackedBiLSTM for the problem of turbine disc fir-tree slot broaching tool wear recognition. Figure 2 shows the StackedBiLSTM structure.

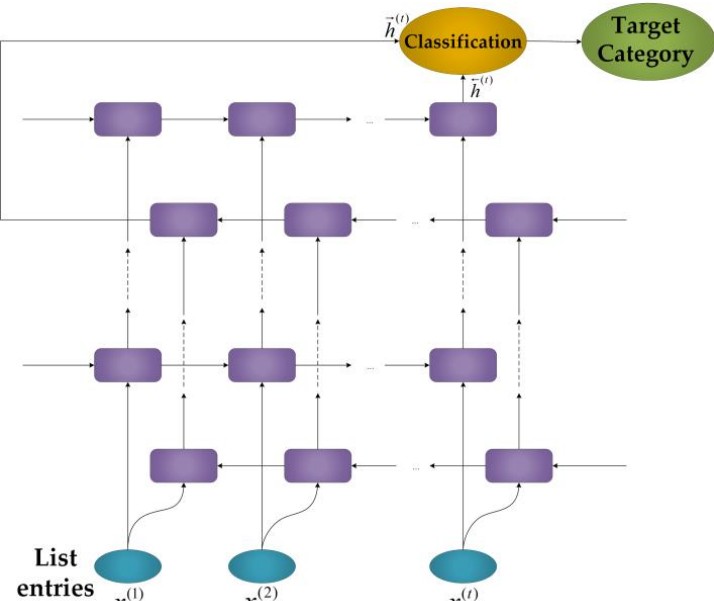

**Figure 2.** The StackedBiLSTM structure.

Therefore, in this paper, a ShuffleNet v2.3-StackedBiLSTM broach tool wear state recognition model is finally proposed for the problem of turbine disc fir-tree slot broach tool wear recognition.

### 2.3. Classification Mechanism

The CNN and RNN are used as feature extractors of vibration signals. However, the fully connected neural network with Softmax should be used to classify the tool wear. Its input and output satisfy a specific mapping relationship. The calculation process is stacked together by fully connected dense layers, and each layer (except the output layer) is connected to the next layer, which is the key of the fully connected layer architecture. The formula is as follows:

$$y_i = f(W_i^T x_i + b_i) \tag{7}$$

where $W_i^T$ and $b_i$ respectively represent the weight matrix and bias matrix, $x_i$ and $y_i$ respectively represent the input and output of layer $i$. The activation function $f()$ is the ReLU function. Finally, the tool wear states are classified by Softmax function.

## 3. Broaching Tool Wear State Recognition Model

### 3.1. Model Construction

Based on the ShuffleNet v2.3-StackedBiLSTM neural network structure, the broaching tool wear state recognition model is established as shown in Figure 3. Table 2 shows the details of each layer of the broaching tool wear state recognition neural network. First, the vibration signal is normalized and then fed into the lightweight ShuffleNet v2.3 neural network to extract spatial features and keep the size output small. Then, the StackedBiLSTM neural network is used to extract the time series information, thus compensating for the lack of only spatial feature information extracted by the ShuffleNet v2.3 network. A maximum pooling layer is added after the StackedBiLSTM network structure to allow the network to reduce parameters and computation while retaining the main feature information, preventing overfitting, and improving the model generalization capability. In addition, two fully connected layers were used to add nonlinear outputs. Finally, the successfully trained high-level form was used with Softmax to estimate the tool wear state.

**Table 2.** Broaching tool state recognition model details.

| Neural Network | Network Layer | Output Layer | Details |
|---|---|---|---|
| Input | - | - | - |
| | Conv-1 | $1 \times 24 \times 15,000$ | Conv1D; 24; Kernel size = 3; Stride = 2 |
| | MaxPool1d | $1 \times 24 \times 7500$ | Kernel size = 3; Stride = 2 |
| | Stage 2 | $1 \times 36 \times 3750$ | |
| | Stage 3 | $1 \times 72 \times 1875$ | Table 1, Figure 2 |
| ShuffleNet v2.3 | Stage 4 | $1 \times 144 \times 938$ | |
| | Conv-2 | $1 \times 576 \times 938$ | Conv1D; 576; Kernel size = 1; Stride = 1 |
| | AvgPool1d | $1 \times 576 \times 938$ | Kernel size = 3; Stride = 1 |
| | Dropout | $1 \times 576 \times 938$ | p = 0.2, inplace = False |
| | BiLSTM | $1 \times 938 \times 128$ | bidirectional = True |
| | BiLSTM | $1 \times 938 \times 256$ | bidirectional = True |
| StackedBiLSTM | MaxPool1d | $1 \times 938 \times 128$ | Kernel size = 3; Stride = 2 |
| | Fc-1 | $1 \times 64$ | in_features = 128, out features = 64 |
| FNN | Linear | $1 \times 3$ | in_features = 64, out features = 3 |
| | Classification | $1 \times 3$ | Softmax(dim = 1) |

Note: Inputs is $1 \times 3 \times 30,000$ (3 represents x, y and z directions); Conv stands for Conv1d-BatchNorm-ReLU; Fc stands for Linear-BatchNorm1d-ReLU-Dropout.

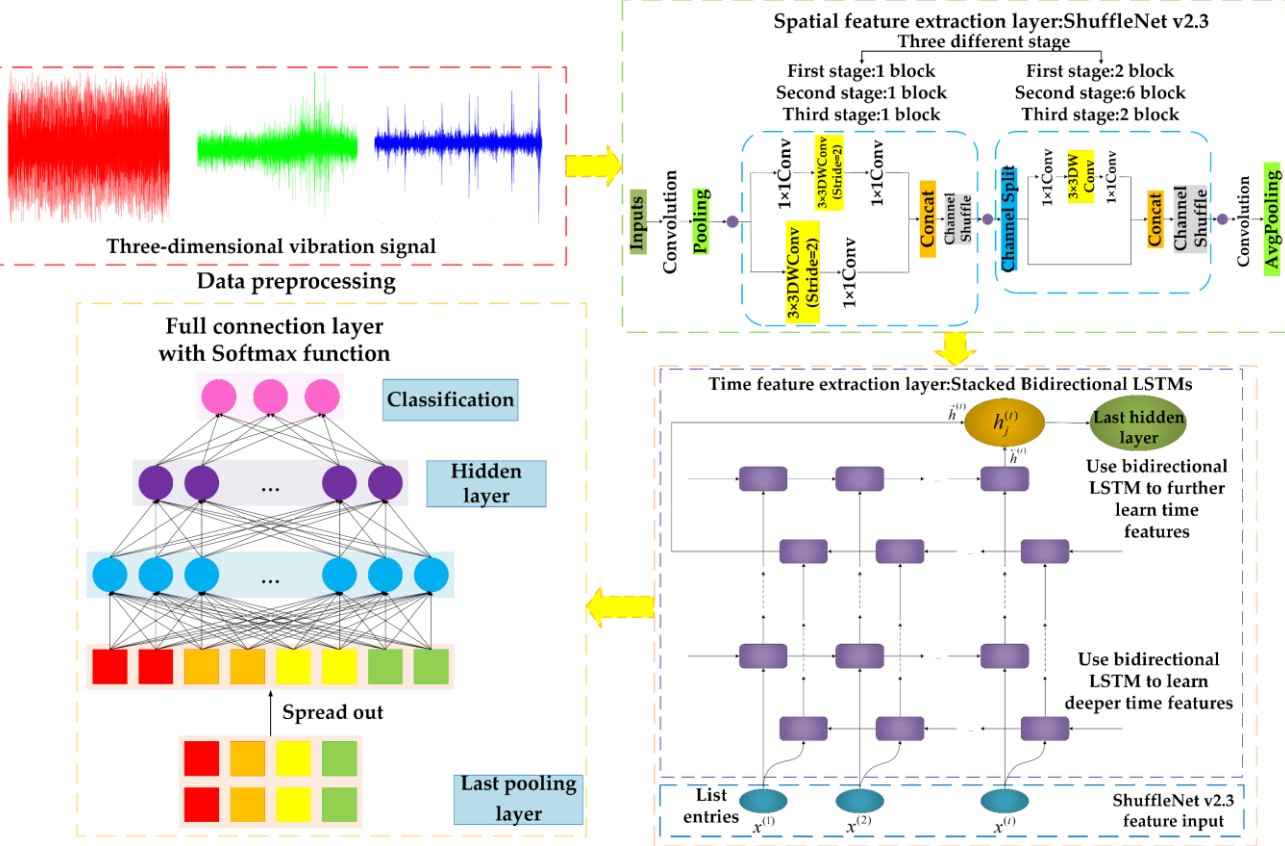

**Figure 3.** ShuffleNet v2.3-StackedBiLSTM tool wear state recognition model.

*3.2. Model Training*

As the collected vibration signals have different value ranges, data normalization is necessary to enable features to have the same metric scale. The original vibration signal is normalized according to Equation (8):

$$\overline{x} = \left| \frac{x}{x_{\max}} \right| \tag{8}$$

where $x_{\max}$ indicates the maximum value of the signal. Both the training and test set signals must be uniformly transformed (normalized) in order not to affect the ability of the signal to characterize the tool wear state, and the signals before and after normalization are shown in Figure 4.

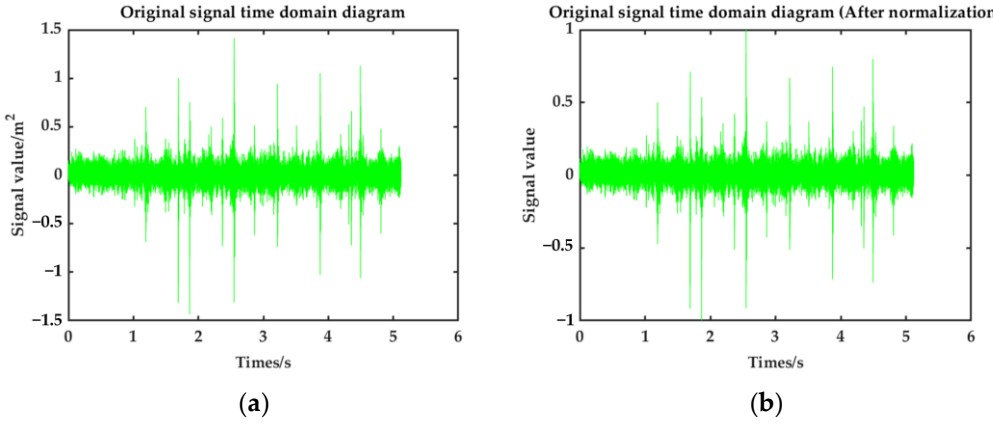

**Figure 4.** Data pre-processing before training: (**a**) Original signal; (**b**) normalized signal.

The model in this paper will use a cross-entropy loss function in the training process for the broaching tool wear state recognition classification problem, as defined in Equation (9):

$$L = -\frac{1}{N}\sum_j \sum_{c=1}^{M} y_{jc} \log(p_{jc}) \tag{9}$$

where $M$ represents the number of categories, $y_{jc}$ represents the sign function (0 or 1), taking 1 if the true category of sample $j$ is equal to $c$ and 0 otherwise, and $p_{jc}$ represents the predicted probability that the observed sample $j$ belongs to category $c$.

The main processes for training and testing the ShuffleNet v2.3-StackedBiLSTM model are as follows:

1.  The original vibration signals were obtained on the fir-tree slot broaching platform, and the original signals were normalized and the vibration signal samples were marked with wear labels. 80% of the vibration signal samples were randomly selected as the training set to train the model, and the remaining 20% were used as the test set to evaluate the trained and optimized model;
2.  The training set was put into the ShuffleNet v2.3-StackedBiLSTM tool wear state model for training, and the model selects the adaptive moment estimation optimization algorithm to perform the weight update using the back propagation technique. When the pre-defined maximum number of iterations or loss value is reached, the weight update is terminated and the training model with optimal parameters is obtained;
3.  The test set was put into the ShuffleNet v2.3-StackedBiLSTM model for evaluation testing.

During the training process, if the loss function does not show a decreasing trend, the model is overfitted and the model structure is adjusted for training. Conversely, the model converges, the model parameters are adjusted until they have a high accuracy, and the model structure and parameters are saved for the industrial process.

## 4. Experimental Validation and Analysis of Results

### 4.1. Eperimental Setup and Procedure

Figure 5 shows the experimental platform. The machine tool is a high-speed horizontal side broaching machine (model: LG6516zx-2800) with the main parameters shown in Table 3. The workpiece is an aircraft turbine disc of a certain type and the workpiece material is the high temperature nickel-based powder alloy FH97. The tool used in this test is a coated carbide broach.

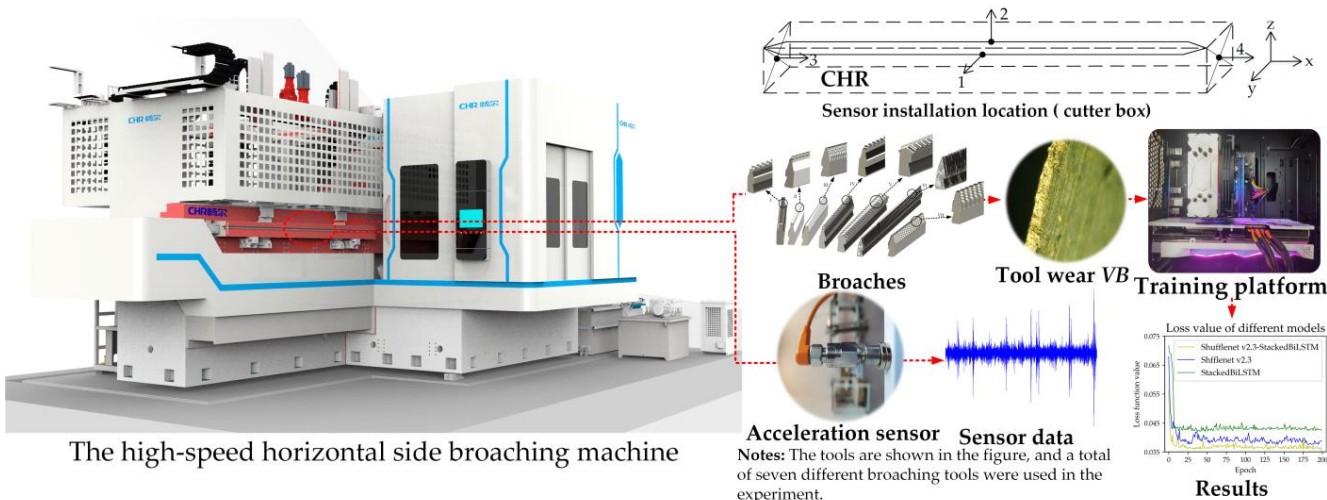

**Figure 5.** Test device diagram of tool wear experiment.

**Table 3.** Main parameters of horizontal broaching machine LG6516zx-2800.

| LG6516zx-2800 Main Parameters | | | |
|---|---|---|---|
| Main motor power | 51 kW | Rated broaching force | 160 kN |
| Broaching stroke | 2800 mm | Broaching speed | 4 m/min |

The vibration signal acquisition system uses the IFM vibration monitor VSE150, 4 channels, communication interface Profinet, and supports Ethernet and OPC. The acceleration sensor adopts IFM capacitive MEMS accelerometer VSA001. The vibration measurement range is −25 g~25 g, the frequency range is 0–6000 Hz, the measurement principle is capacitive, the sensitivity is 142 μA/g.

A high-performance server was used for the experimental training deep learning hardware platform, and the relevant specifications of the server are shown in Table 4.

**Table 4.** Server specifications.

| Server Specifications | |
|---|---|
| Central processing unit | 9th Gen Intel(R) Core(TM) i7-9700 3.00 GHz |
| Memory | Kingston DDR4 32.0 GB |
| Graphics processors | NVIDIA GeForce RTX 2070 Super |
| Operating systems | Windows 10 Enterprise Edition |
| Deep learning frameworks | Pytorch 1.10.2 |
| Unified Computing Architecture | CUDA 11.3 |

The tool wear is measured by digital microscope system: Keyence VHX-970FN, camera lens: VH-Z250R, RZ × 250 −× 2500, contour measuring unit: VHX-S15.

Four vibration pick-up points are arranged in the middle and on both sides of the cutter box, two in the x direction and one each in the y and z directions.

In the experiment, seven broaching tools were used to complete the broaching operation, and a total of 790 sets of original signal samples under different wear conditions of broaching tools were obtained. The fir-tree slot broaching tool processes the workpiece once, which is recorded as a broaching stroke. At the end of each broaching stroke, the fir-tree slot broach is observed and measured by a Keyence camera lens assembled to the tripod head. Measure and record the average tool wear at 1/2 of the tool face after the first tooth of rough, semi-fine, and fine process.

According to the tool wear process and combined with the actual experimental situation, when the rear tool face wear is 0~0.05 mm, the broaching tool wears rapidly in a short period of time, the stage is classified as the initial wear stage. When the rear tool face wear is 0.05~0.2 mm, the broaching tool cutting process is stable, the stage is classified as the middle wear stage. When the rear tool face wear is greater than 0.2 mm, the tool rapidly reaches a failure state, the stage is classified as the severe wear stage. At the same time, the tool wear state is recorded as three data tags, and the tool wear state was coded using a one-bit effective encoding form.

*4.2. Eperimental Setup and Procedure*

To further verify the generalization capability, superiority, and reliability in the ShuffleNet v2.3-StackedBiLSTM tool wear state recognition model, the ShuffleNet v2.3 model and StackedBiLSTM model were experimentally compared with the ShuffleNet v2.3-StackedBiLSTM model. The parameters of the three models were set the same in the process of experimental training. The model specific parameter settings are shown in Table 5.

After the different classification models were trained and tested, different loss values and accuracy rates were obtained, and their loss value and accuracy rate change curves are shown in Figure 6. As can be seen from Figure 6, with the increase of the number of iterations epoch, the loss function value of each model showed a significant decreasing

trend, and the accuracy rate gradually increased and fluctuated in a small range, and no gradient explosion occurred, and the model finally reached convergence.

**Table 5.** Model specific parameters.

| Parameters | Model | | |
|---|---|---|---|
| | ShuffleNet v2.3-StackedBiLSTM | ShuffleNet v2.3 | StackedBiLSTM |
| Learning rates | 0.001 | 0.001 | 0.001 |
| Weight decay | 0.01 | 0.01 | 0.01 |
| Momentum factor | 0.99 | 0.99 | 0.99 |
| Epochs | 200 | 200 | 200 |
| Number of batches | 16 | 16 | 16 |
| Dropout | 0.5 | 0.5 | 0.5 |
| Optimization algorithm | Adam | Adam | Adam |

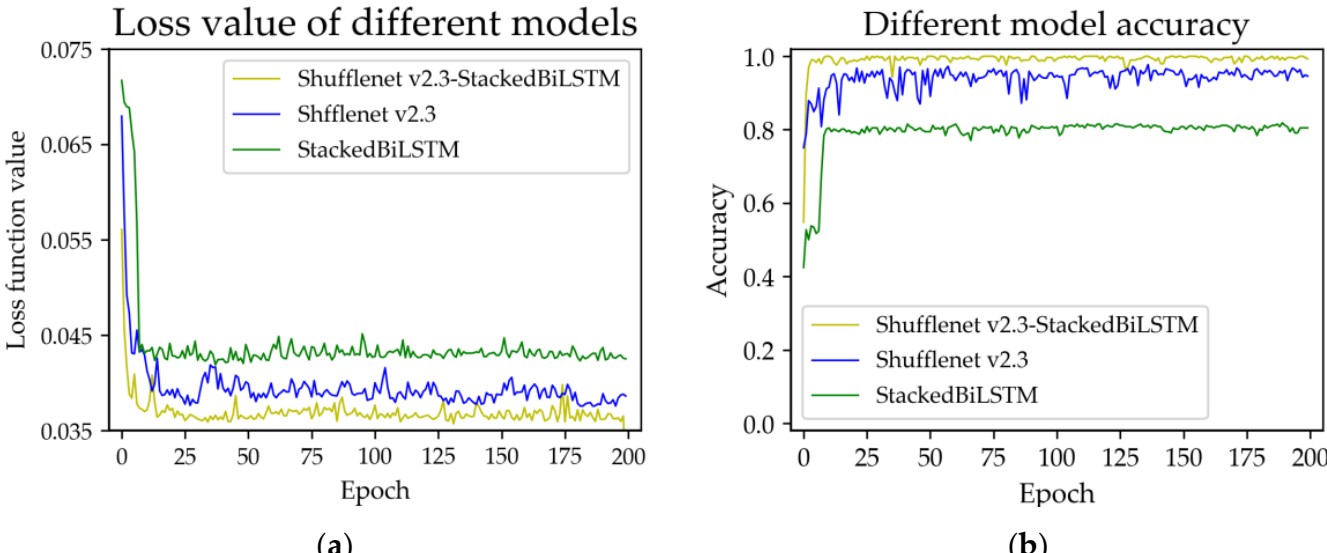

**Figure 6.** Loss value and accuracy change curve of each model: (**a**) Loss value curve; (**b**) accuracy curve.

The total number of test set samples was 158, and the test sets were put into the converged and saved ShuffleNet v2.3-StackedBiLSTM, ShuffleNet v2.3, and StackedBiLSTM models for test evaluation respectively. The results of the test evaluation through the confusion matrix are shown in Figure 7.

From the recognition results in Figure 7 and Table 6, the accuracy of ShuffleNet v2.3 model, which focuses only on spatial feature information, is 94.93%. The accuracy of StackedBiLSTM model, which focuses only on temporal feature information, is 80.37%. Although the ShuffleNet v2.3 model has a low number of parameters, low computational cost, and takes up less memory, the precision rate is 85.71% for severe tool wear and the recall rate is only 89.74% for initial tool wear, which is not enough to meet the requirements of industry. However, the StackedBiLSTM neural network does not have a convolutional layer to reduce the dimension of the original vibration signal, which results in a large number of model parameters and huge computational costs. Moreover, since only the time feature sequence of tool wear state is extracted, both the average precision and the average recall rate are low. In terms of the single precision rate, the tool only achieves 44.12% in the severe wear condition, which can be said that the recognition effect of the model is less than ideal.

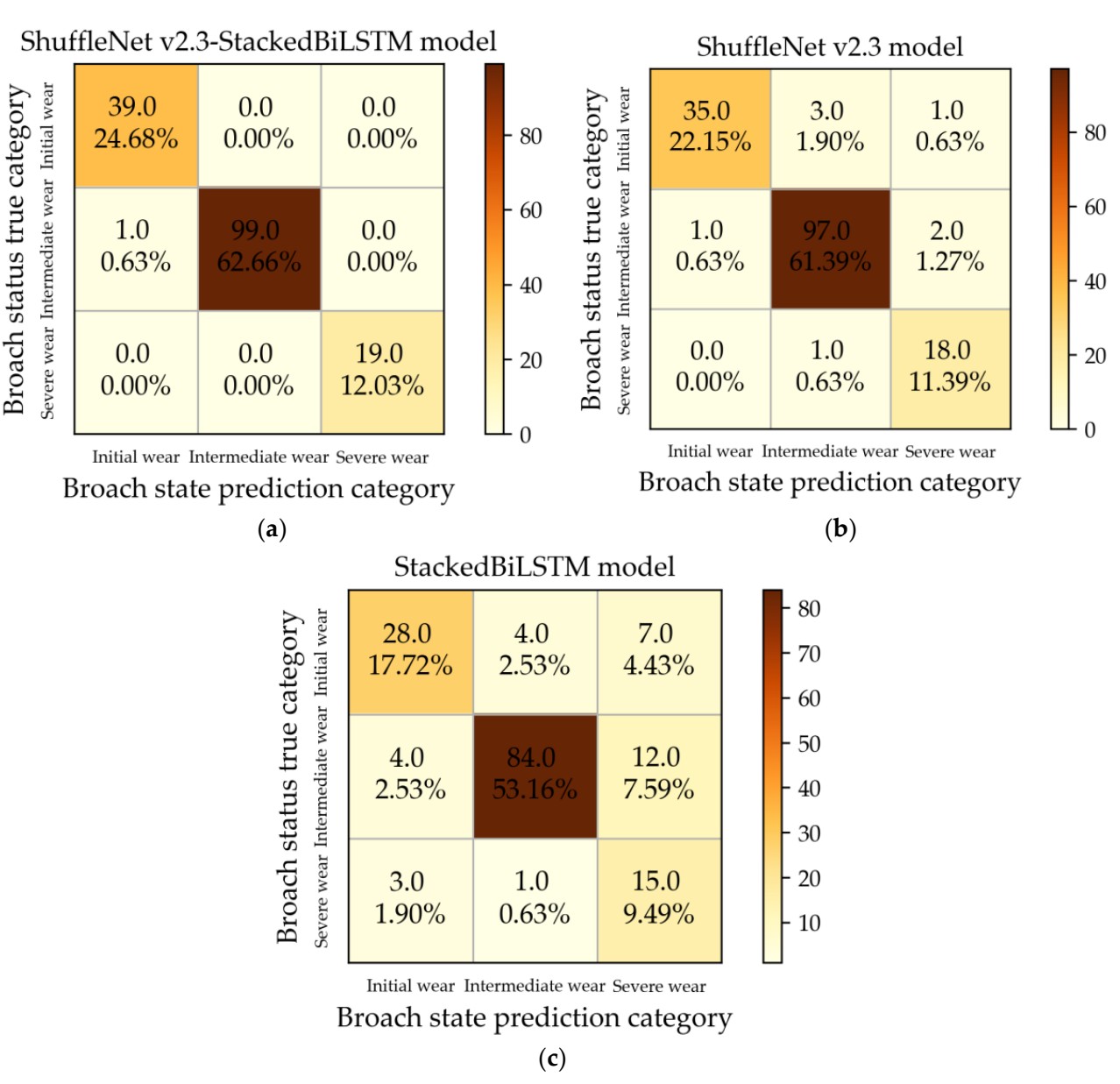

**Figure 7.** Confusion matrix for different models:(**a**) ShuffleNet v2.3-StackedBiLSTM classification model confusion matrix test results; (**b**) ShuffleNet v2.3 classification model confusion matrix test results; (**c**) StackedBiLSTM classification model confusion matrix test results.

**Table 6.** Comparison of experimental results of different models.

| Models | Single Test Time/ms | Accuracy/% | Precision/% | Recall/% | F1 Value/% | Number of Model References/pc |
|---|---|---|---|---|---|---|
| ShuffleNet v2.3-StackedBiLSTM | 8.67 | 99.37 | 99.38 | 99.37 | 99.37 | 1,244,249 |
| ShuffleNet v2.3 | 6.45 | 94.93 | 95.09 | 94.94 | 94.95 | 208,601 |
| StackedBiLSTM | 89.59 | 80.37 | 84.79 | 80.38 | 81.75 | 16,161,155 |

F1 is a comprehensive metric that aims to better balance the impact of precision and recall and evaluate a classifier comprehensively, with a higher value indicating a higher quality model. The ShuffleNet v2.3-StackedBiLSTMm model achieves a good balance of both F1 values and model parametric numbers.

The accuracy of the ShuffleNet v2.3-StackedBiLSTM model was 99.37%, which was 4.44% and 19% higher than ShuffleNet v2.3 and StackedBiLSTM models respectively. This indicates that using both spatial feature extraction and temporal feature extraction can capture deeper hidden features in broaching vibration signals. The ShuffleNet v2.3-StackedBiLSTM model increased the single test time by 2.22 ms compared to the ShuffleNet v2.3 model. Although ShuffleNet v2.3 used channel shuffling to effectively reduces the computational effort of $1 \times 1$ pointwise convolution, making the network extremely lightweight and efficient. However, because of the large number of StackedBiLSTM parameters, the computational inference time increases, thus increasing the single test time. Although StackedBiLSTM model takes advantage of stacked bidirectional to improve the expressive power of the network and alleviate the forgetting problem. However, the StackedBiLSTM model does not extract spatial features, making it unable to meet industrial requirements in terms of both single test time and accuracy. ShuffleNet v2.3 model has the fastest time in a single test, but its accuracy cannot meet the requirements. In summary, the ShuffleNet v2.3-StackedBiLSTM neural network has higher recognition accuracy and shorter recognition time, and is more suitable for industrial applications.

## 5. Conclusions

Because of the special characteristics of turbine disc fir-tree slot broaching tool in geometry and space position, this paper proposes an improved deep learning model combining the advantages of ShuffleNet v2.3 and StackedBiLSTM models to recognize tool wear in broaching. The ShuffleNet v2.3 network is used to mine spatial correlation features of vibration signals, while StackedBiLSTM makes up for the deficiency of spatial features by acquiring temporal correlation features. The experimental results show that ShuffleNet v2.3-StackedBiLSTM neural network has deeper network structure, and its classification accuracy, precision, and recall rate are 4.44%, 4.29%, and 4.43% higher than ShuffleNet v2.3. Compared with StackedBiLSTM, the classification accuracy rate, precision rate and recall rate are increased by 19.00%, 14.59%, and 19.32%, and the single test time is optimized by 90.32%. Almost all the different performance indicators are better than ShuffleNet v2.3 and StackedBiLSTM models, which are more suitable for online recognition and monitoring of tool wear status in industrial sites.

Deep learning requires a large amount of data for model training, and the tool state recognition model also needs to be dynamically adjusted with the change of broaching system parameters. Therefore, the types of sensors will be further increased in the future, more data will be obtained, and more comprehensive state features will be extracted for tool wear state. The model used this time is a lightweight model, which lays a foundation for its deployment to the mobile terminal in the future.

**Author Contributions:** Methodology, Y.S.; software, S.Y., Y.S. and F.Z.; validation, Y.S.; formal analysis, Y.S.; investigation, Y.S.; resources, S.Y. and L.L.; data curation, Y.S.; writing—original draft preparation, Y.S.; writing—review and editing, S.Y.; project administration, S.Y.; funding acquisition, S.Y. All authors have read and agreed to the published version of the manuscript.

**Funding:** This research work was supported by the Zhejiang Province Welfare Technology Applied Research Project (Grant No. LGG21E050017).

**Institutional Review Board Statement:** Not applicable.

**Informed Consent Statement:** Not applicable.

**Data Availability Statement:** Not applicable.

**Conflicts of Interest:** The authors declare no conflict of interest.

## Nomenclature

| | |
|---|---|
| $\sigma$ | Sigmoid function |
| $W_i$ | Cyclic weight matrix |
| $W_f$ | Cyclic weight matrix |
| $W_c$ | Cyclic weight matrix |
| $W_o$ | Cyclic weight matrix |
| $b_i$ | Bias |
| $b_f$ | Bias |
| $b_c$ | Bias |
| $b_o$ | Bias |
| $h_{t-1}$ | Previous moment hidden layer state |
| $x_t$ | Current moment input |
| $\widetilde{C}_t$ | Temporary unit state |
| $C_t$ | The unit state at the current time |
| $i_t$ | Input gate |
| $f_t$ | Forget gate |
| $o_t$ | Output gate |
| $W_i^T$ | Weight matrix |
| $b_i$ | Bias matrix |
| $x_i$ | Input of layer $i$ |
| $y_i$ | Output of layer $i$ |
| $\overline{x}$ | Normalization |
| $x_{\max}$ | Maximum value of vibration signal |
| $y_{jc}$ | The sign function (0 or 1) |
| $M$ | The number of categories |
| $p_{jc}$ | The predicted probability |
| $j$ | The observed sample |
| $C$ | Category |

## Abbreviations

| | |
|---|---|
| RUL | Remaining Useful Life |
| AE | Acoustic Emission |
| SVM | Support Vector Machine |
| RF | Random Forest |
| PNN | Probabilistic Neural Network |
| GMHMM | Gaussian Mixed Hidden Markov Model |
| HSMM | Hidden Semi-Markov Model |
| BPNN | Back Propagation Neural Network |
| LS-SVM | Least Square Support Vector Machine |
| PCA | Principal Component Analysis |
| CNN | Convolutional Neural Network |
| RNN | Recurrent Neural Network |
| DBN | Deep Belief Network |
| LSTM | Long Short Term Memory |
| SVR | Support Vector Regression |
| ANN | Artificial Neural Network |

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
