# Peer review of "ShuffleNet v2.3-StackedBiLSTM-Based Tool Wear Recognition Model for Turbine Disc Fir-Tree Slot Broaching"

_machines, doi:10.3390/machines11010092_

Round 1
Reviewer 1 Report
1. It is suggested to put Section 2.1 into the introduction and modify the introduction.
2. Delete the introductory language in Section 2.2.
3. Figure 6 is not clear enough.
4. Three line tables are recommended for all tables.
5. The conclusion needs to be rewritten.
Reviewer 2 Report
The authors analyzed the application of various neural networks, in order to recognize the wear of fir-tree sloat broaching tool. It was shown that the ShuffleNet v2.3-StackedBiLSTM neural 357 network has a deeper network structure and outperforms both ShuffleNet v2.3 and 358 StackedBiLSTM models in terms of different performance indicators.
The research is well designed and presented clearly. The methodological section of the manuscript is presented in sufficient detail. However, some minor shortcomings should be corrected to make the manuscript acceptable for publication in Machines.
1. lines 70-71: "The introduction section reviews the current the research status of cutting tool wear state recognition at home and abroad." The literature review in this section is very limited. The introduction needs to be revised completely. The authors directly started from the literature survey by introducing the topic of research. Moreover, the literature survey case studies are not details: the authors just considered mentioning the titles or one-sentence outcomes of the case studies. More details of each discussed study with specific achievements with values should be mentioned in the introduction section.
2. A good comparative analysis of existing publications concerning the tasks set in the work is not performed.
3. Providing the geometry of the tool used would allow for a better interpretation of the results by the readers.
Reviewer 3 Report
This study presents a broaching tool wear recognition model based on ShuffleNet v2.3- StackedBiLSTM. The experimental research and results show that the model has outstanding in accuracy, precision, recall, and F1 value, and the accuracy rate reaches 99.37%, which is far better than ShuffleNet v2.3 and StackedBiLSTM models. The validity of the proposed method of a broaching process is demonstrated. The main idea of this study is interesting, and the contribution is significant. It can be considered to be published with the minor revision to address the following points:
(1) It is necessary for the authors to explicitly give the experiment setup and configuration, especially the sensors installation details. Figure 6 isn’t enough to illustrate the location of sensor installation.
(2) The quality of Lots of the figures are poor to understand the details. For example, figure 6, figure 4,
(3) The inputs of the deep learning neural network are not clearly presented in the whole paper.
(4) In the introduction section, the authors should explain what is main motivation of introducing the new ShuffleNet v2.3- StackedBiLSTM models? In other word, what is the unsolved problem or shortcomings of the existing methods?
(5) In line 291, the authors states that “the tool wear state is recorded as three data tags and the tool wear state is coded using the one-hot code form.” What is the one-hot code form? Please explain it in a easier understanding way.
(6) In line 13 of the abstract, “a broaching tool recognition model …is proposed” should be revised as “a broaching tool wear recognition model …is proposed” as the title indicated.
(7) For readers' convenience, a nomenclature and a list of abbreviation should be added.
Reviewer 4 Report
Dear Authors,
The article I reviewed: “ShuffleNet v2.3-StackedBiLSTM based tool wear recognition model for turbine disc fir-tree slot broaching” takes up the important topic of manufacturing machine parts with complex shapes and high precision. The article is well written, but it has some shortcomings and needs to be corrected. The shortcomings of the article include:
- Page 1, line 29 - The authors use the term "high tolerance" - in my opinion it is rather a narrow field of tolerance, while "high tolerance" can be understood as great tolerance = wide tolerance.
- Page 1, line 32 - is: "manufacturing[1].", should be "manufacturing [1]." - space was missing. This error is repeated throughout the article.
- Page 10, figure 8 - the middle field is too dark. This makes it difficult to read the numerical value entered here. A lighter color will improve the readability of the drawing.
- Page 11, line 317-341 - the description of the obtained results is very imprecise. The authors begin this fragment with the words: "From the recognition results in Figure 8 and Table 6 ...", meanwhile in the text there are numerical values that have no reference to either Figure 8 or Table 6: line 321 - "the precision rate is 85.71% " , line 322 - "recall rate is only 89.74%", line 327 - "precision rate, the tool only achieves 44.12%", etc. If the authors quote specific values of the tested parameters, they must be verifiable. This needs to be improved.
- Please reconsider your conclusions. The conclusions in their current form contain general information about the conducted research. Maybe it's worth shortening the conclusions but writing them more specifically, citing the measurement values ...?? It's about pointing out specific benefits.
- The selection of literature is correct. Only a few items are older than 5 years.
Please consider my comments and make the necessary corrections.
In my opinion, these changes will increase the quality and readability of the work.
Best regards,
Reviewer
Round 2
Reviewer 1 Report
The reviewer agrees to accept the manuscript as the author has revised it in accordance with the reviewer's opinion.